# Three Months of Strength Training Changes the Gene Expression of Inflammation-Related Genes in PBMC of Older Women: A Randomized Controlled Trial

**DOI:** 10.3390/cells11030531

**Published:** 2022-02-03

**Authors:** Keliane Liberman, Rose Njemini, Louis Nuvagah Forti, Wilfried Cools, Florence Debacq-Chainiaux, Ron Kooijman, Ingo Beyer, Ivan Bautmans

**Affiliations:** 1Frailty in Ageing Research Group (FRIA), Gerontology Department, Vrije Universiteit Brussel (VUB), B-1090 Brussels, Belgium; keliane.liberman@vub.be (K.L.); Rose.Njemini@vub.be (R.N.); fortilouis@yahoo.fr (L.N.F.); ingo.beyer@vub.be (I.B.); 2Interfaculty Center Data Processing and Statistics (ICDS), Vrije Universiteit Brussel (VUB), B-1090 Brussels, Belgium; Wilfried.cools@vub.be; 3URBC, NAmur Research Institute for LIfe Science (NARILIS), University of Namur, B-5000 Namur, Belgium; florence.chainiaux@unamur.be; 4Center for Neurosciences (C4N), Vrije Universiteit Brussel, B-1090 Brussels, Belgium; ron.kooijman@vub.be; 5Geriatrics Department, Universitair Ziekenhuis Brussel, B-1090 Brussels, Belgium

**Keywords:** resistance training, aged, inflammation, gene expression, leukocytes

## Abstract

Here, we investigate changes in inflammation-related gene-expression in peripheral mononuclear blood cells (PBMC) by strength training. A total of 14 women aged ≥65 years were randomized into 3 months of either 3×/week intensive strength training (IST: 3×10 rep at 80% 1RM), strength endurance training (SET: 2×30 reps at 40% 1RM) or control (CON: 3×30 sec stretching). Differentially expressed genes (fold change ≤0.67 or ≥1.5) were identified by targeted RNA-sequencing of 407 inflammation-related genes. A total of 98 genes (*n* = 61 pro-inflammatory) were significantly affected. IST and SET altered 14 genes in a similar direction and 19 genes in the opposite direction. Compared to CON, IST changed the expression of 6 genes in the same direction, and 17 genes in the SET. Likewise, 18 and 13 genes were oppositely expressed for, respectively, IST and SET compared to CON. Changes in gene expression affected 33 canonical pathways related to chronic inflammation. None of the altered pathways overlapped between IST and SET. Liver X Receptor/Retinoid X Receptor Activation (LXR/RXR) and Triggering Receptor Expressed On Myeloid Cells 1 (TREM1) pathways were enriched oppositely in both training groups. We conclude that three months IST and SET can induce changes in CLIP-related gene expression in PBMC, but by affecting different genes and related pathways.

## 1. Introduction

Ageing is accompanied by elevated levels of circulating inflammatory biomarkers, described as a chronic low grade inflammatory profile (CLIP) [1]. Exercise is one of the most effective non-pharmacological means to counter CLIP and has shown to have anti-inflammatory effects [2,3]. Studies have shown that between 6 and 12 weeks of resistance training significantly reduced the basal circulating IL6 levels, thus reflecting a decreased level of CLIP [4,5,6]. Exercise provokes an acute and brief liberation of myokines such as IL6 from the contracting muscle, which induce an anti-inflammatory response, probably by, respectively, inhibiting and stimulating the production of pro- and anti-inflammatory cytokines by peripheral blood mononuclear cells (PBMC) [6,7]. Exercise-induced changes in IL-6 have been extensively studied, mainly for prolonged exercise (>2 h running, bicycling or resistance exercise, see Ref [8] for review). However, shorter exercise bouts at submaximal intensity [9,10,11] and short repeated bouts at maximal intensity [12,13,14] also elicit an acute increase in circulating IL-6. Importantly, the exercise-induced myokine production is maintained at a higher age and a regular strength training session is sufficient for obtaining an average increase of ~20% in circulating IL-6 in older adults (aged 60–80 years) [4]. Circulating IL-6 increases during acute exercise, reaching a maximum near the end of the session and returning to baseline values within 24–48 h [13,15,16,17,18,19]. This myokine response is different from the cytokine release seen during inflammation. In fact, during inflammation (e.g., acute infection), TNF-α and IL-1β are the first cytokines to appear in the circulation. Next, IL-6 increases, followed by IL-1ra, sTNF-R and IL-10, which control and limit the inflammatory reaction. The exercise-induced acute elevation in IL-6 is not preceded by increased TNF-α levels [15] but is immediately followed by significant elevations in IL-1ra and sTNF-R [15] (inhibiting, respectively, IL-1β and TNF-α) and the anti-inflammatory cytokine IL-10, thus reducing inflammation [15]. Here, IL-6 is believed to have inflammation-reducing effects. A mechanism by which exercise could decrease CLIP is through the anti-inflammatory response of PBMC by the stimulation of receptors on their cell surface [20,21,22]. Although this effect is washed out after approximately 24 h [23], there is growing evidence that long-term repeated bouts of exercise can switch the basal pro-inflammatory secretion profile of PBMC in older adults to an anti-inflammatory one [2,19,24].

Myokines secreted through muscle contraction have been shown to stimulate the acute secretion of cytokines by PBMC in the circulation. However, currently there is little experimental evidence on whether an exercise intervention can change the basal inflammatory expression profile of circulating PBMC in older persons. In young, trained men, 30 min of strength training at moderate intensity (45–65% of 1 repetition maximum (RM)) was sufficient for upregulating inflammation-related genes in PBMC (AREG, DUSP2, NR4A2, CREM, EREG and RGS1) [25]. Similarly, in healthy young adults (aged 23–45 years), an acute bout of exhaustive exercise affected the expression of many genes in PBMC including HIF1a, HSP, CCL3, CCL4, CD69, IL6R, XCL1, MYC and EREG [26,27]. Jimenez-Jimenez et al. investigated the change in expression of inflammatory-related genes in PBMC of older men (66–75 years) following an acute bout of eccentric leg exercise both before and after an 8-week eccentric leg training program [28]. In the untrained condition, the acute bout of eccentric training increased the expression of COX2, IL6 and iNOS. After 8 weeks of training, these acute exercise-induced increases were strongly attenuated, pointing to a long-term adaptation of the inflammatory gene expression in PBMC.

Compared to young adults, the PBMC of older adults showed a higher expression of RAGE, NF-Kb, iNOS, TNF-alpha, MCP-1 and other inflammatory related genes. After 2 months of aerobic training, the expression of these genes was attenuated in the older participants [29]. In the same line, 6 months of aerobic exercise training reduced the expression of pro-inflammatory genes and increased the expression of anti-inflammatory ones in PBMC of older adults with mild cognitive impairment [30]. Moreover, 8 weeks of strength training decreased TNF-alpha gene expression in PBMC of older women [31]. This reduction in TNF-alpha expression was accompanied by a significant decrease in circulating levels of TNF-alpha, IL6 and CRP, reflecting lower CLIP.

Physical exercise is now considered a powerful means to prevent and counter chronic diseases [32,33]. Many different exercise interventions have been proposed to counter CLIP in older adults but no consensus yet exists regarding the most effective exercise modality [34]. Given the age-related loss of muscle mass and muscle strength in ageing, defined as sarcopenia [35], there is increasing interest in strength training programs for older persons as these are considered the most effective means to combat sarcopenia [35,36,37,38]. Since CLIP is considered as an aggravating factor for sarcopenia [39], an exercise intervention able to counter both conditions might be a preferred intervention for prevention and treatment. In a recent review, Borde et al. showed that training period, intensity, and total time the muscles are under tension play an important role for increasing muscle strength in older adults [40]. Regarding the exercise-induced anti-inflammatory effects, we previously showed that there is a dose-response relationship between the intensity of strength training and the effects on CLIP. Furthermore, 12 weeks of strength training at high external loads (80% of 1RM) increased circulating IL1Ra in older men while training at lower loads (20% to 40% of 1RM) did not alter IL1Ra levels [41]. Additionally, in a similar study in younger adults we showed that although the beneficial effects of 9 weeks strength training on IL6 and IL8 were not dependent on exercise intensity, the levels of sTNFR1 increased after training at higher loads but not at lower loads [19].

The underlying mechanisms for these dose-response relationships remain unclear. It can be hypothesized that different intracellular pathways are activated in PBMC following strength training at different intensities. Understanding how strength training intensity influences the anti-inflammatory exercise-induced effects is crucial for exercise prescription to counter CLIP in older persons. Therefore, the aim of this study was to investigate the changes in expression of inflammation-related genes in PBMC of older women after 3 months of different modalities of strength training compared to control. 

## 2. Materials and Methods

### 2.1. Participants & Randomization

The SPRINT study (Senior PRoject INtensive Training) is an ongoing randomized controlled study in which community-dwelling older adults aged ≥65 years are included for 3 to 6 months of resistance training at different intensities to evaluate their immune system. The detailed inclusion criteria and protocol have been previously described [42]. Briefly, participants were recruited through flyers distributed at the geriatric day hospital of the UZ Brussel, and by general practitioners and senior associations in the Brussels environment. Participants were excluded if they presented contra-indications for the exercise interventions or if they were unable to understand or execute the exercise instructions due to cognitive impairment (mini mental state examination (MMSE) < 24/30) [43] or physical disability. Although comorbidities were not an exclusion criterion per se, participants showing acute uncontrolled conditions and/or acute inflammation (CRP ≥ 10 mg/L) were excluded. Subjects performing currently or within the past 6 months, on a regular basis, physical exercise at higher intensities than habitual daily activity (e.g., fitness classes, strengthening exercises, cycling club) were excluded. Included participants were randomly assigned to either Intensive Strength Training (IST), Strength Endurance Training (SET) or Control (CON). Randomization was performed by a blinded researcher and was stratified by age (65–74/≥75 years), gender and health category (based on a modified SENIEURS’s protocol, described by our group previously [44]). Physical activity level was measured using the Yale Physical Activity Survey (YPAS) questionnaire and the Activity Dimensions Summary score (YPAS-ADS) was calculated, reflecting the subject’s physical activity (vigorous activity, leisure walking, moving, standing, and sitting) over the last month on a scale from 0 (no activity at all) to 177 (maximal activity) [45].

For this sub-study, similarly as in our previously published paper [42], we included only women as the target sample size for the female participants had been reached. Women who completed 3 months of intervention, did not become ill during the training period, did not change their medication intake for at least 6 weeks prior blood sampling, were sufficiently compliant to the intervention, and of which the RNA samples had an RNA Quality Number (RQN) higher than 7 and a yield higher than 8 ng/µL were eligible for further analysis. 

The study protocol was approved by the local ethics committee in accordance with the Declaration of Helsinki and each participant provided written informed consent prior to medical examination (RIB approval codes 2005/104 and 2011/257).

### 2.2. Intervention

The training intervention has previously been described by Cao Dinh et al. [42]. Briefly, all of the training sessions took place at the exercise facilities of the Brussels Health Campus of the Vrije Universiteit Brussel on Technogym™ (Technogym, Gambettola, Italy) and Matrix^®^ (Matrix, Cottage Grove, WI, USA) single station cable-type devices and were supervised by trained experts to minimize the risk of injury and ensure the participants used the proper technique and weights and performed the exercise throughout the entire range of motion. 

Each training session, regardless of the allocated training group, started with a warm-up of 10 exercises that included mobility and activating exercises for both the upper and lower limbs (such as circular movements with the limbs, but no treadmill or cyclo-ergometer), intended to prepare the joints and muscles for the upcoming exercises. Each warm-up exercise was performed without external resistance for a total of 15 repetitions per extremity or per exercise, the duration of the complete warm-up was approximately 5 to 10 min. 

The IST and SET groups performed six exercises (seated chest press, seated leg press, seated hip abduction, seated hip adduction, seated low row, and seated vertical traction) consisting of intensive sub-maximal muscle contractions with both concentric and eccentric components. There was no predetermined sequence of the 6 exercises within each training session, however, participants were instructed to alter exercises for upper and lower limbs. Participants were asked to exercise 2–3 times weekly, with a minimum of a 1-day interval for recovery. The strength training protocols were designed to be approximately equal in volume (% 1RM multiplied by the number of sets and repetitions). The IST group performed 3 sets of 10 consecutive repetitions at 80% of 1RM. For the SET group, the exercises were designed similarly as for the IST group, but with less intensive muscle contractions and a higher number of repetitions (2 sets of 30 consecutive repetitions at 40% of 1RM). The rest between sets was minimum 1 min for both IST and SET groups. Both strength training programs entailed an initial accommodation period of 2 weeks in which the target exercise intensity was progressively reached. Every 2 weeks, the individuals’ 1RM for each training device was determined and exercise loads were adapted accordingly. A minimum of 24 h of rest (i.e., no training) was scheduled before and after a 1RM test. For the 1RM determination the participant started with a warming up of 20 repetitions at 30% of the estimated or previously assessed 1RM. Next, the participant was asked to perform 1 repetition at 70% of the estimated or previously assessed 1RM. This step was repeated with increasing load until the participant is unable to perform the exercise correctly in full range of motion. The 1RM was considered as the highest load at which the participant was able to perform the exercise correctly in full range of motion. The load corresponding to 1RM was reached in a maximum of 4 to 5 steps.

The CON group performed a “placebo” flexibility training consisting of 3 sets of sustained (30 s) passive, static stretching exercises of the large muscle groups. Stretching exercises principally act by applying mechanical tension on the muscles and tendons, leading to improved range of motion [46]. Therefore, this type of exercise, which mainly induces a passive load on the muscles and tendons without muscle contractions or cardiovascular challenge, was chosen as a control intervention.

The muscle strength index (MSI) was calculated at baseline and after 3 months of training as the average of the 1RM for the 6 strength training exercises. Mean exercise compliance was calculated as follows: participants were asked to train 3×/week for 3 months resulting in a target number of 36 sessions. The effectively performed number of training sessions was recorded and was expressed as a percentage of the target number.

### 2.3. Blood Cell Counts, PBMC Isolation and RNA Extraction

As previously described [42], peripheral EDTA blood was collected before (at baseline) and after 3 months of intervention, at least 24 h after the last exercise session. The blood samples were first exposed to lysis buffer for 10 min for lysation of the red blood cells. Next, the leucocytes were centrifuged at 900 g for 4 min at room temperature. These cells were then isolated and washed twice in PBS containing 1% BSA at 900× *g* for 3 min and re-suspended in 200 µL PBS containing 1% BSA. The samples were then analysed using a Coulter FC 500 flow cytometer (Beckman Coulter, Fullerton, CA, USA). Data acquisition was performed using the Coulter CXP software (Epics, version 2.3). The leukocyte subpopulation was analysed through haematology analyses and gated according to size and granularity in the forward versus side scattergram, thereby excluding dead cells. Lymphocyte subsets were quantified as previously described [42]. Antibodies were initially titrated to determine the optimal conditions for flow cytometry analysis before staining. Approximately 5 × 105 cells were stained with 3 μL each of PE-CY5-labelled anti-CD8 (Becton Dickinson, San Jose, CA, USA), PE-CY7-labelled anti-CD3 (Biolegend, San Diego, CA, USA), FITC-labelled anti-CD28 (Biolegend, San Diego, CA, USA), Dazzel-labelled anti-CD45 and PE-labelled anti-CD57 (Biolegend, San Diego, CA, USA). After 20 min of incubation at room temperature in the dark, the cells were washed at 900× *g* for 3 min and 500 μL of FACS flow solution (Becton Dickinson, San Jose, CA, USA) were added. The lymphocyte subpopulation was gated according to size and granularity in the forward vs. side scattergram, thereby excluding dead cells. Fluorescence-minus-one controls were used to distinguish positive from negative events and the various lymphocyte clusters were identified according to their expression of a combination of surface markers. The expression (or non-expression) of CD28 and CD57 are particularly useful in distinguishing between subsets of differentiated T-cells. Based on these surface markers, CD8^–^ and CD8^+^ T-cells were separated into four distinct sub-populations including CD28^+^CD57^–^, CD28^–^CD57^–^, CD28^–^CD57^+^ and CD28^+^ CD57^+^. We used the terminologies naive (CD28^+^CD57^–^, consisting predominantly of naive T-cells and perhaps some early differentiated T-cells), memory (CD28^–^CD57^–^), and senescence-prone (CD28^–^CD57^+^ and CD28^+^CD57^+^) phenotypes to define the distinct subsets as previously described [42]. Absolute blood counts were measured using a dual platform methodology (flow cytometry and the Cell-Dyn Sapphire haematology analyser (Abbott Diagnostics Division, Santa Clara, CA, USA)).

PBMC were isolated using Lymphoprep (density gradient: 1.077 ± 0.001 g/mL, Axis-Shield, Oslo, Norway) density gradient centrifugation. Total RNA was collected through purification of RNA using the GeneJet RNA purification kit (Thermo Fisher Scientific, Waltham, MA, USA) according to the manufacturer’s protocol. Purified RNA was then immediately stored at −80 °C until simultaneously assayed for all of the participants at once.

### 2.4. RNA Quality Assessment & RNA Sequencing

Prior to analyses, a selection of genes which had been shown to be involved in inflammation in young or older adults, or in CLIP, or which had previously been reported to be modified at least 1.3-fold after an exercise intervention were included in our analyses. Based on these criteria, a set of 407 genes with their corresponding customed probes were identified and included in this RNA sequencing experiment. 

All of the analyses were performed by the BrightCore facility of the UZ Brussel. The quality of the total RNA samples was assessed on the AATI Fragment Analyzer (Agilent Technologies Inc., Santa Clara, CA, USA), using the DNF-472 High Sensitivity RNA Analysis Kit. Included RNA samples had an RNA Quality Number (RQN) higher than 7 and a yield higher than 8 ng/µL.

RNA libraries were created from 150 ng of total RNA per sample, using the KAPA RNA HyperPrep Kit with RiboErase kit (Roche Diagnostics, Vilvoorde, Belgium) according to the manufacturer’s instruction. Briefly, following ribodepletion and DNase digestion, the RNA was fragmented to average sizes of 200–300 bp by incubating the samples during 6 min at 94 °C. Following first strand synthesis, second strand synthesis and adapter ligation, the libraries were amplified using 12 PCR cycles. Next, 12 libraries were pooled by equal mass (83.3 ng each) for a total of 1 µg of cDNA library and captured according to the Roche SeqCap RNA Enrichment System User’s Guide v1.1 (Roche Diagnostics, Vilvoorde, Belgium), with 2 modifications: (1) the usage of xGen Universal Blockers TS Mix (Integrated DNA Technologies, Coralville, Iowa, USA) in the capturing reaction and (2) limiting the PCR to 12 PCR cycles. Final libraries were qualified on the AATI Fragment Analyzer (Agilent Technologies Inc., Santa Clara, CA, USA), using the DNF-474 High Sensitivity NGS Fragment Analysis Kit and quantified on the Qubit 2.0 with the Qubit dsDNA HS Assay Kit (Life Technologies, Carlsbad, CA, USA). Per sample, 25 million 2 × 100 bp reads were generated on the Illumina NovaSeq 6000 system (Illumina Inc., San Diego, CA, USA), with the NovaSeq 6000 S2 Reagent Kit (200 cycles) kit. For this, 1.9 nM libraries were denatured according to manufacturer’s instruction. STAR (v 1.5) was used for alignment and htseq-count (version 0.11.0) to obtain the counts.

### 2.5. Statistical Analyses & Pathway Analyses

Differences in baseline characteristics between the three intervention groups were analysed using a one-way ANOVA with LSD post-hoc analyses. Changes in MSI were analysed through a repeated-measures ANOVA with the MSI as within subject and intervention group as between subject factors. As blood counts were mostly not-normally distributed, non-parametric tests were used, more specifically the Kruskal-wallis test for baseline differences between groups (Mann-Whitney U test for post-hoc analyses) and the Wilcoxon-signed ranked test for changes after the intervention. All of the above mentioned analyses were performed using IBM SPSS^®^ version 25.0.

Differentially expressed genes were identified using DESeq2 version 1.22.2 [47] in Rstudio version 1.1.463 (Boston, MA, USA). Power calculation using Gpower^®^ [48] indicated that a sample size of 4 participants per group enabled a detection of an exercise-induced gene expression fold change of at least 1.5 with a power of 0.82 and alpha = 0.017. After normalization of the count files, the log_2_ fold change (FC) was calculated for each group separately. The log_2_FC was then back-transformed to obtain an absolute FC for all of the genes. All of the genes with an absolute FC of ≤0.67 or ≥1.5 were considered as clinically relevant. All of the genes occurring with a clinically relevant FC in at least one of the intervention groups were pooled and compared with the corresponding FC values of the other intervention groups (Figure 1, Figure 2, Figure 3 and Figure 4). Next, these genes were categorized to either pro-inflammatory genes (Figure 2), anti-inflammatory genes (Figure 3) or genes of which the function of inflammation is not yet fully determined (Figure 4), according to their known function based on gene databases from the National Center for Biotechnology Information (Gene [Internet]. Bethesda (MD): National Library of Medicine (US), National Center for Biotechnology Information; 2004– [cited in 6 July 2019]. Available from: https://www.ncbi.nlm.nih.gov/gene/) as well as from previously published literature including these genes. 

Pathway analysis was performed with Ingenuity Pathway Analyses (IPA) on all of the genes from our RNAseq experiment. Tissue types were filtered including only immune cells and immune cell lines and excluded cancer-related and xenobiotic pathways. Pathways with a Benjamini-Hochberg *p*-value < 0.05 and a z-score of ≤−2 or ≥2 were considered as significant. Z-scores were calculated using IPA taking into account the activation or inactivation of the genes based on the fold changes, as well as the causal relationships between these genes. Pathway analyses revealed activation or inactivation of 42 pathways across all of the intervention groups, of which 33 were relevant to the immune system and related to exercise or CLIP (Figure 5). Pathways such as sperm motility, acute myeloid leukaemia signalling, endometrial cancer signalling were considered as irrelevant and excluded. These pathways were further analysed trough Ingenuity Target Explorer^®^ (https://targetexplorer.ingenuity.com/index.htm accessed on 8 July 2019). The main genes involved in the activation or inactivation of these pathways were listed with their respective fold changes. Although the expression was not significantly affected for all of the genes after the intervention, their interaction with other genes might have led to the enrichment of these pathways.

## 3. Results

A total of 14 women were included in the analyses, 4 from the IST group, 5 from the SET group and 5 from the CON group. The IST group showed a slightly but statistically significant higher percentage of monocytes compared to the SET. There were no other significant differences between the 3 intervention groups at baseline nor regarding exercise compliance (Table 1). 

As shown in Figure 1, the overall muscle strength, expressed as MSI, showed a significant time (*p* = 0.024) and time × training interaction effect (*p* = 0.029). Muscle strength of the IST group improved from 24.3 ± 6.1 kg to 33.1 ± 8.5 kg (significant increase of 35%, *p* = 0.005) and the SET from 34.5 ± 8.7 kg to 45.7 ± 7.9 kg (significant increase of 33%, *p* = 0.002). The CON group showed no significant change in MSI (from 37.6 ± 14.5kg to 34.2 ± 4.9, decrease of 9%, *p* = 0.54). The total number of WBC and T-cells, as well as the proportions of WBC and T-cell subtypes remained unchanged after the 3 months intervention (Table 1). 

### 3.1. Exercise-Induced Gene Expression Changes in PBMC

A total of 98 genes with a significant FC after intervention were identified (Figure 2, Figure 3, Figure 4 and Figure 5), of which 61 are considered as pro-inflammatory, 28 as anti-inflammatory and 9 of which the pro- or anti-inflammatory role is not yet fully described in the context of exercise. 

#### 3.1.1. Pro-Inflammatory Genes

For the three intervention groups, different expression patterns were noticed. Six genes (IL1A, CXCL1, IL1B, CXCL2 and CCL3) were downregulated in the IST while they were upregulated in both the SET and CON. On the other hand, 2 genes (LTB4R2_A and KLK15) showed opposite expression patterns. Moreover, 2 genes, CD68 and RGS1 were both downregulated in the IST and SET, while they were upregulated in the CON. The opposite was seen for CPA2, that was upregulated in the IST and SET, while it was downregulated in the CON. Another 2 genes (BDKRB1 and THY1) were only downregulated in the SET while upregulation was seen in IST and CON. CXCL10 was upregulated in IST and CON, but no changes were seen in SET. In both the IST and SET, NCAM1 and COL3A4 were upregulated and REN was downregulated, while no changes were seen in the CON. In the IST, CCL2 and PLA2G5 were downregulated, while no changes were seen in the SET and upregulation was seen in the CON. Nine genes were altered in the SET and CON but not in the IST. IL17A, LRP2 and BDKRB2 were upregulated in the SET and downregulated in CON. The opposite was seen for TERC and CXCL11. CCL4 was upregulated while DNM1P46, HIF3A and SLC2A4 were downregulated in both groups. In the IST only, 6 genes were upregulated (CD160, KLK3, LTC4s, PTGER3, CRISP3, CCR3) and 5 genes were downregulated (COL1A1, LTB4R, NR4A2, EPHB2 and IL5). In the SET only, 9 genes were upregulated (PDE4C, IL1R2, TREM1, CSF3R, DUSP2, COL4A1, CXCR1, BMX, MMP9) and 2 genes were downregulated (PRKAA2, IL3). In the CON group only, 7 genes were all upregulated (KLK2, IL6, HTR3B, PLA2G2D, CD69, PLA2G2A, TNFSF10).

#### 3.1.2. Anti-Inflammatory Genes

In all of the groups, IL1RAPL2 was upregulated, while FRZB was downregulated. For IL-9 and NFKBIA, downregulation was seen in the IST while upregulation was seen in the SET and CON groups. Both the IST and SET upregulated the expression of MXRA5 and HSPA1A. Additionally, for MXRA5, downregulation was also found in the CON. In the IST, FOXP3 and PTGIS were upregulated while these genes were downregulated in the SET and no change was seen in the CON. While in the IST, no changes were observed in expression of FCGR3B, in the SET and CON this gene was upregulated. In the IST only, 4 genes (HRH1, FGF21, IL4 and ARNT2) were upregulated, while 1 gene (ASPN) was downregulated. In the SET only, 7 genes (THBD, HSPA1B, SOD2, HSPA1L, CA4, HSP6 and CSF3) were upregulated and 5 genes (AGTR1, CSF2, IGF2, IL2 and GFG6) were downregulated. Last, for the CON only, upregulation of SOC3 was found.

#### 3.1.3. Genes of Which the Role in Inflammation Has Not Yet Been Fully Determined

Upregulation of CYP1A2 and MYH6 was seen in both the IST and SET while for the control, downregulation was seen. In both IST and SET, downregulation was seen for PRM1 and RPL3L while no changes were observed for CON. In both the IST and CON, CACNA2D1 was upregulated while it was downregulated in the SET. In the IST only, PTGFR was upregulated. In the SET only, 2 genes (DAAM2 and CYP7A1) were upregulated. In the CON, only OLIG2 was downregulated.

### 3.2. Pathway Analyses

Figure 6 shows the canonical pathways that were significantly altered by IST, SET and/or CON. In Appendix A for each of the affected pathways, the involved genes are represented with their respective up- or down-regulation. Those altered after the exercise intervention are discussed in relation to their relative affected pathway.

“TREM1 signalling” was inhibited in IST (z-score = −2.04), while it was equally activated in the SET (z-score = 2.41) and CON (z-score = 2.41) groups. This pathway is mediated by a transmembrane adaptor molecule DNAX-activating protein 12 (DAP12) and leads to cell adhesion as well as the activation of the adaptive and innate immune system. It also reduces the Th2 response and increases TLR signalling and Th1 response. Finally, this pathway increases proinflammatory immune responses through among others enhanced TLR signalling. Several pro-inflammatory genes play important roles in this pathway, such as IL6, IL1B, MCP1 (also known as CCL2), CCL3, MIP2 (known as CXCL2) and TREM1, which showed all a (tendency for) downregulation in the IST but upregulation in the 2 other groups.

“LXR/RXR” was activated in the IST (z-score = 2.04) but blunted in the SET (z-score = −2.04), group, while no changes were seen in the CON (z-score = −0.93) group. This pathway has been shown to have (anti-)inflammatory and anti-angiogenic effects through secretion of pro-inflammatory mediators such as LXR, which has been shown to be a negative regulator of macrophage inflammatory gene expression [49]. IL1RAPL2 plays a very important role in this pathway and was upregulated in all 3 groups, though the highest upregulation was seen in the IST (FC = 20.52) compared to the SET (FC = 6.94) and the CON (FC = 5.54). The difference in how IST and SET affected this pathway could rely on the changes in the expression of CCL2 on one hand, which was downregulated in the IST (FC = 0.42) while it was upregulated in the SET (FC = 1.31), and on the other hand of PTGS2, showing inverse expression where in the IST it was downregulated and in the SET it was upregulated (FC = 0.58 and FC = 4.22, respectively). Additionally, IL1A and IL1B were downregulated in the IST (FC = 0.15 and 0.26) and upregulated in the SET (FC = 2.27 and 2.31).

The pathway “cytotoxic T lymphocyte-mediated apoptosis of target cells” was activated in the IST (z-score = 2.12) and CON (z-score = 2.12) groups, while no significant changes were seen in the SET (z-score = −1.41) group. These differences were mainly induced by the increased expression of PRF1, CD3D, FAS and BCL2 in the IST (respectively, FC = 1.25; 1.32; 1.01 and 1.11) and CON (FC = 1.18; 1.09; 1.14 and 0.92), while a decreased expression was seen in the SET (FC = 0.93; 0.95; 1.01 and 0.90). Besides CD3D, other members of the CD3 family play a role in the activation of this pathway, namely CD3E and CD3G (FC = 1.15 and 1.17 for the IST and FC = 1.06 and 1.04 for the CON, respectively).

Nine pathways were exclusively modulated after IST, two of which were upregulated and seven were downregulated. The 2 pathways that were activated where the IL7 signalling pathway and the IL22 signalling pathway. The IL7 pathway leads to anti-apoptotic and proliferative effects by binding to the IL7 receptor, and many genes involved in this process were affected in the IST, such as STAT1 (FC = 1.11), MAPK3 (FC = 1.10), JAK3 (FC = 1.14), STAT5 (FC = 1.08) and IFNG (FC = 0.78). Concerning the IL22 pathway, similar genes as in the IL7 pathway led to the activation of the IL22 pathway while SOC3 was downregulated (FC = 0.92) decreasing the negative feedback control of IL22 signalling. The “dendritic cell maturation” pathway, mainly activated through cytokines, microbes, and immune cells, was the most strongly inhibited pathway in the IST group (z-score = −2.941) while this was not significantly altered in the other groups (SET: z-score = 0.70; CON: z-score = −0.14). The “maturation” part of this pathway was inhibited mainly through the downregulation of IL1A (FC = 0.15), IL1B (FC = 0.26), IL6 (FC = 0.71), TLR3 (FC = 1.09) and TLR4 (FC = 0.79). Downregulation of IL6 and IL12A (FC = 0.85) decreased the “Th1 and Th2 polarization” part of the pathway. The “activation of dendritic cells and Th1 development” part of the pathway is strongly mediated by IL1B and STAT4 (FC = 0.94), which were both downregulated in the IST group. The second most inhibited pathway in the IST group was the “aryl hydrocarbon receptor signalling” pathway (z-score = −2.84), which mediates a wide range of toxic responses, apoptosis and cell proliferation. Key regulators for inactivation of this pathway in the IST group were IL1A, IL1B and IL6. Increased expression ARNT (FC = 1.01), leading to increases in expression of BAX (FC = 1.05), Fas (FC = 1.04) and FasL (FC = 1.19) lead to apoptosis through this pathway. Additionally, ARF, known as CDKN2A (FC = 1.20) was also upregulated in this pathway, leading to secretion of proteins involved in cell cycle regulation and apoptosis. 

The third most inhibited pathway was the “role of pattern recognition receptors of bacteria and viruses” pathway (z-score = −2.24). Activation of this pathway results in the secretion of pro-inflammatory cytokines. In the IST group, this pathway was blunted mainly by the downregulation of IL1A, IL1B, IL6, TLR7 (FC = 0.98), TLR4 (FC = 0.79), TLR6 (FC = 0.80), TLR2 (FC = 0.77), IL5 (FC = 0.66), IL2 (FC = 0.74), IL3 (FC = 0.86), IL18 (FC = 0.84) and IL12A (FC = 0.85). The “Toll-like receptor signalling” pathway, responsible for the transcription of pro-inflammatory cytokines [50], was strongly blunted in the IST group (z-score = −2.18). Two genes, namely IL1A and IL1B, play important roles in the inactivation of the “apoptosis result” part of this pathway, which were downregulated in the IST. Additionally, NFKBIA (FC = 0.51) plays a mediating role in the transcription of pro-inflammatory cytokines within this pathway, such as IL1, TNF-alpha and IL12. This latter gene was also downregulated in the IST group. “Macrophage migration inhibitory factor (MIF) regulation of innate immunity”, a potent pro-inflammatory pathway, was inhibited in the IST (z-score = −2.14), while it remained unchanged in the SET (z-score = 0.54) and CON (z-score = 1.07). MIF modulates the innate immune response through modulation of toll-like receptor 4, and prevents apoptosis of macrophages through inhibition of p53, a tumor suppressor protein [51]. In the IST, mainly PTGS2 (FC = 0.58), NFKBIA, TLR4, CD14 (FC = 0.97) and PLA2G5 (FC = 0.48) lead to inactivation of this pathway.

In the SET group, 6 pathways were affected after the intervention, of which 4 were activated and 2 blunted. The “neuroinflammation signalling” pathway (z-score = 3.05), involved in the destruction and removal of damaged cells, was activated in the SET group through increased expression of CX3CL (FC = 5.71), PTGS2 (FC =4.22), IL1B (FC = 2.31) and MMP9 (FC = 2.23), as well as through decreased expression of PLA2G2D (FC = 0.71), IL12A (FC = 0.76), IFNG (FC = 0.81) and TLR3 (FC = 0.81). Changes in ageing- and exercise-linked genes in the “osteoarthritis” pathway (z-score = 2.48), were limited. Only TGFb (FC = 1.07) and AMPK (FC = 0.26) were affected, though these changes were not sufficient to activate this pathway. While FRZB, important in the progression of osteoarthritis, was decreased (FC = 0.26), mainly the inflammatory factors related to osteoarthritis such as IL1R (FC = 1.24), IL-1b (FC = 2.31), MMP9 and PTGS2 showed an increased expression after the intervention. In the “ErbB2-ErbB3 signalling” pathway (z-score = 2.33), leading to cell division, apoptosis, cell motility and cell adhesion, the MAPK3 (FC = 1.13), MAP2K2 (FC = 1.06), STAT3 (FC = 1.05), STAT5A (FC = 1.02) and STAT5B (FC = 1.03) were increased in the expression after 3 months of SET. The “nuclear factor erythroid 2–related factor 2 mediated oxidative stress response” pathway (z-score = 2.12) was activated in the SET, mainly due to increases in expression of SOD2 (FC = 1.68), ACTB (FC = 1.18), CAT (FC = 1.08) and MAPK3. Activation of this pathway through these genes leads to reduction in oxidative damage. The “sirtuin signalling” pathway that was inhibited in SET (z-score = −2.52) is a very large and complicated pathway, involved in many functions. In the SET, increases in NOS2 (FC = 1.08), stimulated by NFkB (FC = 0.97), can lead to increased inflammation, while decreases in SOD1 (FC = 0.97) lead to decreased production of ROS. Although HIF3a (FC = 0.22) was decreased after the intervention, the expression of SOD2 was increased after the exercise intervention leading to increased oxidative stress. Additionally, BAX (FC = 1.00) and STAT3 were increased which led to increased apoptosis and Th17 differentiation, respectively. Last, the “OX40 signaling” pathway (z-score = −2.71), responsible for T lymphocyte expansion, was inhibited in the SET group. This inhibition was characterised by a decreased expression of IL2 (FC = 0.59), MAPK8 (FC = 0.91) and NFKB1 (FC = 0.97). In addition, the decreased expression of BCL2L1 (FC = 0.90) and BCL2 (FC = 0.90) lead to the reduction in cell survival in this pathway.

As can be seen in Figure 5, in the CON group, 11 pathways were activated, and 4 pathways were blunted. In many of these pathways, the members of the phospholipase C family were involved of which the expression was often decreased in the CON group, leading to the inactivation of these pathways.

## 4. Discussion

In this study, we investigated the impact of 3 months of strength training at either high-load (IST) or low-load (SET), compared to a stretching control (CON) intervention on the change in expression of inflammation-related genes in PBMC (Figure 7). Overall muscle strength significantly increased over time in the IST and SET groups (respectively, +35% and +33%), indicating that our training intervention was sufficient in eliciting physiological adaptations. A total of 98 genes showed significant changes in expression. Among these, 89 were significantly affected by the strength training intervention. Strikingly, IST and SET altered only 14 genes in similar direction whereas 19 genes were altered in opposite direction. Although the strength gains were comparable, many differences in gene-expression between IST and SET were found. Compared to CON, IST induced changes in expression of 6 genes in the same direction, while this was the case for 17 genes in the SET group. Likewise, the expression of 18 and 15 genes was in opposite direction for, respectively, IST and SET compared to CON. This is reflected by the fact that none of the enriched pathways overlapped between both strength training modalities. In addition, 2 pathways were enriched oppositely. Interestingly, many of the affected pathways are related to the Senescence Associated Secretory Profile (SASP), such as TREM1, LXR/RXR, Toll-Like Receptor Signalling, T-cell Exhaustion Signalling, Telomerase Signalling and Th2 pathways. This supports the hypothesis that strength training can affect and/or attenuate SASP, resulting in lower CLIP [52,53,54,55].

In the 98 significantly altered genes, mostly pro-inflammatory genes (*n* = 61) showed changes in expression after the intervention. This indicates that exercise might lead to an altered expression of pro-inflammatory genes in PBMC in combination with an increase in anti-inflammatory genes. The effects on anti-inflammatory genes showed mixed results, though most genes were upregulated in the IST and SET. In the CON, some genes were upregulated though most genes did not change in expression after the intervention. For genes of which the exact pro- or anti-inflammatory effect is unclear, mixed results were found, although in the IST and SET, most of these genes showed upregulation while almost no differences were seen in the CON.

To our knowledge, this is the first study investigating the changes in inflammation-related gene expression in PBMC after two modalities of prolonged strength training in older adults. As most of the existing research has focused either on young subjects or on acute bouts of exercise, or on gene expression in muscle biopsies, we aimed to investigate the changes in basal expression patterns of the PBMC. Moreover, we performed a broader assay of inflammation-related genes instead of focusing on the limited number of genes whose response to exercise have already been described. 

Exercise-induced changes in IL6 have been widely investigated, showing decreases in both its gene expression as its circulating levels after a prolonged exercise intervention [5]. In our study, IL6 gene expression was upregulated in the CON, thus suggesting the exercise interventions might attenuate these increases. Thompson et al. investigated the effects of 6 months of aerobic exercise in middle-aged men. In participants with high baseline circulating levels of IL6, 53 probes were differentially expressed in PBMC by training, among which IL4, IL8, IL2 that were upregulated and IL6, which was downregulated [56]. 

IL1A and IL1B, both mediating pro-inflammatory cytokines, were downregulated after 3 months of IST, while they were upregulated in the SET group and even more intensively upregulated in the CON. This is in line with our previous findings on exercise-induced changes in CLIP, where only strength training at the high external load, a training modality corresponding with the IST training in the present study, influenced serum IL1ra levels compared to lower intensity training. This suggests that muscle contractions at high intensity might be one of the most important triggers enabling changes in CLIP [41]. 

CCL3 and CCL4, also known as MIP-1a and MIP-1b, are involved in acute neutrophilic inflammation and synthesis of IL1 and TNF-a [57]. For CCL2, CCL3 and CCL4, upregulations were found in the CON, while lower expression was seen after the IST intervention for CCL3 and tendencies to decrease were found for CCL4, suggesting the beneficial effects of strength training on CLIP. CCL2, also known as MCP1 and involved in chemotactic activity for monocytes and basophils, was downregulated in the IST, did not change in the SET and was strongly upregulated in the CON. This was comparable to the studies performed by Yakeu et al. and Gano et al., both, however, using aerobic interventions [29,58].

Previous studies have shown contrasting results concerning changes in expression of HSP after exercise interventions. While Fehrenbach et al. found increases in HSPBP1 after exercise in trained athletes [59], this was not found in another study by Maltseva et al. [60]. This latter study, as well as Sakharov et al. both found increases in HSPA1A after a short-term exercise intervention [61]. In a study previously performed by our group, HSP70 protein level in PBMC was decreased after 6 weeks of intensive strength training in older adults [4]. Similarly, in another study, we showed a significant decrease in extracellular HSP70 in older women following 12 weeks intensive strength training [62]. In the present study we found only a significant upregulation in HSPA1A after IST, whereas several other HSP genes were significantly upregulated in the SET group (besides HSPA1A also HSPA1B, HSPA1L and HSPA6).

As was previously described by Mangine et al., the exercise intensity rather than the total exercise volume might be more important for inducing changes in not only muscle strength but also in certain blood parameters [63]. Antunes et al. confirmed this finding and showed that circulating levels of inflammatory markers such as TNF-a and IL6 were elevated in the group with the higher intensity of exercise [64].

Several genes were found to be differently expressed between the groups following 3 months of intervention. Of these, KLK15 and LTB4R2_A were upregulated in the IST while they were downregulated in the other groups. KLK15 is a peptidase and involved in the regulation of cancer cell growth, migration of cancer cells and immune regulation [65]. The LTB4R2 gene is a pro-inflammatory lipid mediator that is only overexpressed under stress-induced inflammatory conditions [66] and plays an important role in inflammatory diseases such as cancer and asthma [67]. Although the receptor 2A was upregulated, the LTB4R gene was strongly downregulated in the IST group. The inflammatory gene CX3CL1, involved in attraction and migration of monocytes to inflammatory sites [68], was downregulated in the IST and upregulated in the SET and CON. This gene was also found to be downregulated in healthy young adults 24 h after running a half-marathon [69]. 

Moreover, 2 genes, MXRA5 and MYH6 were upregulated in both strength training intervention groups and downregulated in the CON group. MYH6 is mainly expressed in cardiac cells and plays a role in cardiac muscle contraction [70]. TGF-b1 has recently been found to upregulate MXRA5, which has its role as an anti-inflammatory and anti-fibrotic molecule [71]. The upregulation of both these genes has not previously been reported in association with exercise interventions and may have a potential role in the regulation of the anti-inflammatory effects of exercise.

Only in the SET was downregulation of THY1 and CACNA2D1, seen while upregulation was seen in the 2 other groups. The role of THY1 in exercise-induced immune modulation has not been established yet. A study has found that THY1 was downregulated in muscles of young adults after 5 days of bed rest [72]. Both results indicate a possible link with physical (in)activity. CACNA2D1 is a subunit of the voltage-dependent calcium channels which regulate the activation and inactivation of calcium channels and therefore plays an important role in the excitation-contraction coupling. Mutations in this gene have been found to be linked to cardiac diseases such as Brugada syndrome [73]. However, its role in exercise-induced changes in PBMC has not yet been elucidated.

In the CON group, many of the pro-inflammatory genes were upregulated (23 out of the 56), such as IL6, CCL2, CCL3, CCL4, IL1A and IL1B, which might reflect a progressive increase in CLIP. Interestingly, 8 of these pro-inflammatory genes were downregulated and 12 were unaltered in the IST group. This observation was less pronounced in the SET group. This points towards the potential of strength training, especially at a high intensity, to attenuate the progression of CLIP.

This study has some weaknesses. Firstly, as we included only older women, it remains unclear whether similar effects are to be expected in men. As was shown in a study by Abbasi et al., differences in gene expression between men and women might occur, possibly related to sex-related differences in body composition [69]. The main genes involved in the activation or inactivation of the significantly enriched pathways were listed with their respective fold changes. Although the expression was not significantly affected for all genes after the intervention, their interaction with other genes might have led to an enrichment of these pathways. 

Secondly, since many factors can influence inflammatory gene expression in PBMC, it cannot be excluded that our results might be biased by confounding factors. However, we carefully verified that all of the participants did not present with acute diseases during the training period, did not change their medication intake for at least 6 weeks prior to blood sampling, and were sufficiently compliant to the intervention. Additionally, by including an active control group we were able to control for effects not related to the exercise intervention per se, such as visiting the exercise facilities 3×/week. On the other hand, CON group performed stretching exercises, and it cannot be excluded that some changes in gene expression might have been influenced by this type of exercise. 

Previously, we found that 6 weeks SET significantly decreased the proportion of senescence-prone T cells in older women [42]. However, in the present study neither white blood cell and T-cell counts, nor the proportions of their subsets changed after the 3 months interventions. This indicates that the changes in gene expression assessed in the heterologous population of PBMC are probably unrelated to changes in proportions of specific blood cells. There was, however, a small but statistically significant difference in the proportion of monocytes between IST and SET at baseline. It cannot be excluded that this difference at baseline, although proportions remained unchanged after 3 months in all three groups, might have influenced some differences in changes in gene expression between IST and SET. However, the contrasts between IST and CON as well as SET and CON remain unaffected by this potential confounder. Unfortunately, no additional determinations were performed for B-cell and NK-cell subpopulations neither for single cell RNA sequencing. Future studies performing similar analyses on PBMC subsets after cell sorting might help elucidating which cell types are (most) responsible for the observed exercise-induced changes. Although this is a weakness of this study, it must be acknowledged that we performed RNA-sequencing including a set of 407 genes that were a-priori selected based on an extensive literature and database search for all of the relevant genes which had been shown to be involved in inflammation in young or older adults, or in chronic low-grade inflammation, or which had previously been reported to be modified at least by a 1.3-fold after an exercise intervention. Moreover, the sample size for this analysis was a-priori determined to detect a relevant exercise-induced gene expression change with a power of 0.82 and alpha = 0.017. This, together with the randomized controlled study design, are unique assets of this study. To the best of our knowledge, such a large RNA-sequencing of PBMC following different exercise interventions in a randomized controlled experimental design has never been published yet and provides an excellent set of 98 core-genes to be targeted by e.g., exercise immunology researchers in their future studies.

## 5. Conclusions

In summary, this is one of the first studies investigating the benefits of 3 months of different modalities of strength training on changes in expression of inflammation-related genes in PBMCs of older adults. Both IST and SET, although affecting different inflammation-related pathways in PBMC, showed beneficial effects on CLIP-related genes. Our findings suggest that after 3 months of IST, pro-inflammatory pathways are blunted due to downregulation of inflammatory genes in circulating PBMCs. Only 14.3% of the relevantly altered genes were affected in the same direction by IST and SET, whereas 19.4% were affected in opposite direction. This was reflected by the fact that none of the enriched pathways overlapped between both strength training modalities, while 2 pathways were oppositely enriched. We conclude that 3 months strength training at high and at moderate external load can both induce changes in CLIP-related gene expression in PBMC, but by affecting different genes and related pathways.

## Figures and Tables

**Figure 1 cells-11-00531-f001:**
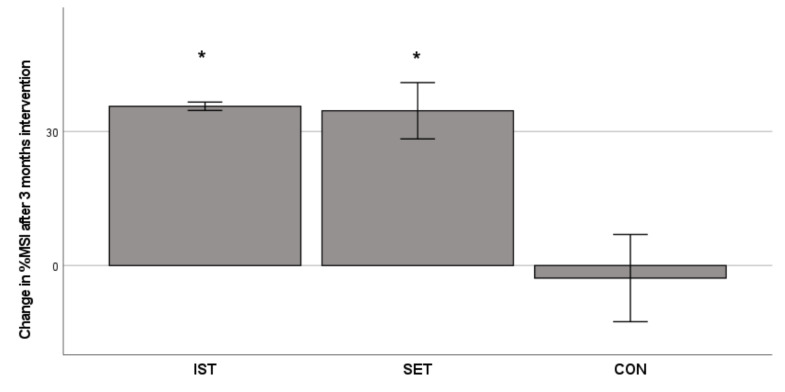
Changes in muscle strength index (MSI, expressed as % change from baseline) after 3 months intervention. Bars represent mean ± SE; * significant change *p* < 0.01.

**Figure 2 cells-11-00531-f002:**
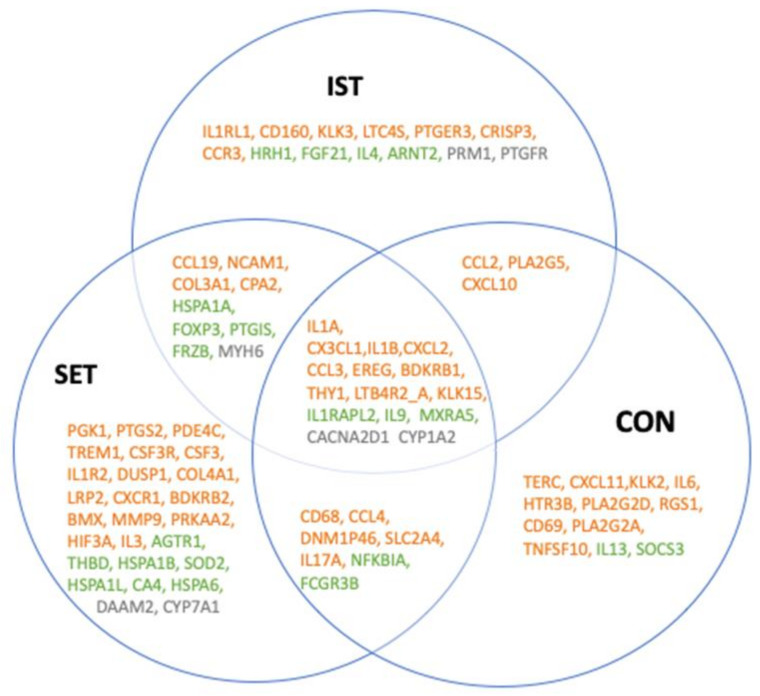
Venn diagram showing the overlap between groups of the 98 genes that were expressed significantly after the intervention. Orange: pro-inflammatory genes. Green: anti-inflammatory genes. Grey: genes where the role in inflammation has not yet been fully determined. Full gene abbreviations list for the can be found in Figure 3, Figure 4 and Figure 5.

**Figure 3 cells-11-00531-f003:**
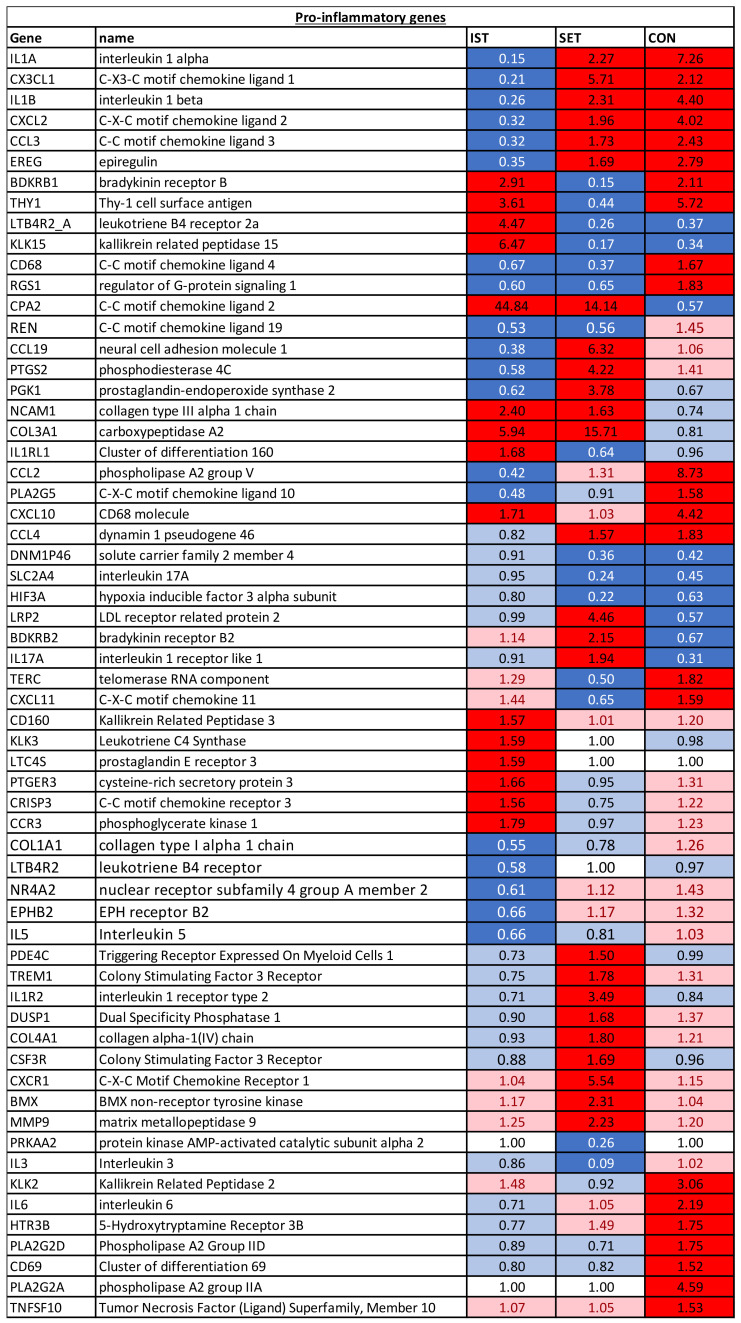
Pro-inflammatory genes of which the expression was changed significantly in at least one of the groups. FC of the genes represent the change from baseline to 3 months intervention. Dark blue: genes that were downregulated after the intervention (FC of ≤0.67). Light blue: genes showing a tendency towards downregulation after the intervention (1 > FC < 0.67). White: genes of which expression was not altered after the intervention (FC = 1); Pink: genes showing a tendency towards upregulation after the intervention (1 < FC < 1.5). Red: genes that were upregulated after the intervention (FC ≥ 1.5).

**Figure 4 cells-11-00531-f004:**
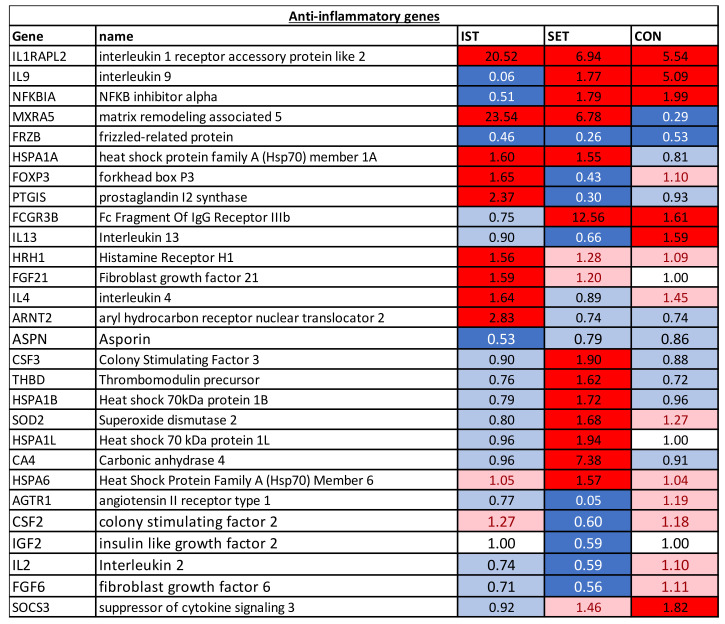
Anti-inflammatory genes of which the expression was changed significantly in at least one of the groups. FC of the genes represent the change from baseline to 3 months intervention. Dark blue: genes that were downregulated after the intervention (FC of ≤0.67). Light blue: genes showing a tendency towards downregulation after the intervention (1 > FC < 0.67). White: genes of which expression was not altered after the intervention (FC = 1); Pink: genes showing a tendency towards upregulation after the intervention (1 < FC < 1.5). Red: genes that were upregulated after the intervention (FC ≥ 1.5).

**Figure 5 cells-11-00531-f005:**
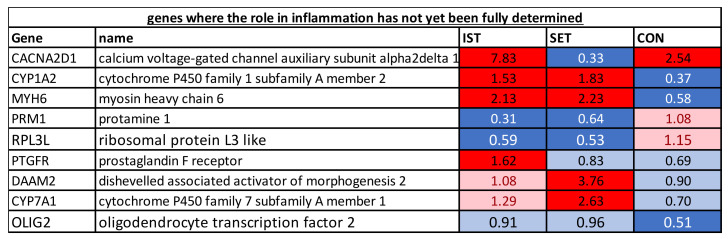
The genes of which the exact immune-modulatory role in the context of exercise has not yet been described and of which the expression was changed significantly in at least one of the groups. FC of the genes represent the change from baseline to 3 months intervention. Dark blue: genes that were downregulated after the intervention (FC of ≤0.67). Light blue: genes showing a tendency towards downregulation after the intervention (1 > FC < 0.67). White: genes of which expression was not altered after the intervention (FC = 1); Pink: genes showing a tendency towards upregulation after the intervention (1 < FC < 1.5). Red: genes that were upregulated after the intervention (FC ≥ 1.5).

**Figure 6 cells-11-00531-f006:**
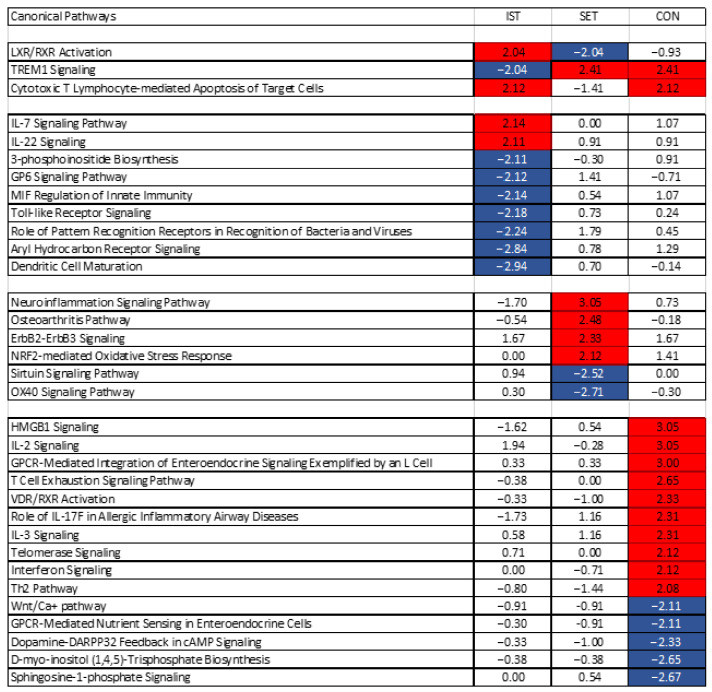
Heatmap of the significantly enriched pathways across the intervention groups. Blue: pathways that were significantly blunted after the intervention (z-score ≤ −2). Red: Pathways that were significantly activated after the intervention (Z-score ≥ 2). LXR/RXR = Liver X Receptor/Retinoid X Receptor, TREM = Triggering Receptor Expressed On Myeloid Cells, IL = Interleukin, GP = Glycoprotein, MIF = Macrophage migration Inhibitory Factor, OX40 = CD134, HMG B = High Mobility Group Box, GPCR = G-protein coupled receptor, VDR = Vitamin D receptor, Wnt = Winless/integrated, DARPP = dopamine- and cAMP-regulated neuronal phosphoprotein.

**Figure 7 cells-11-00531-f007:**
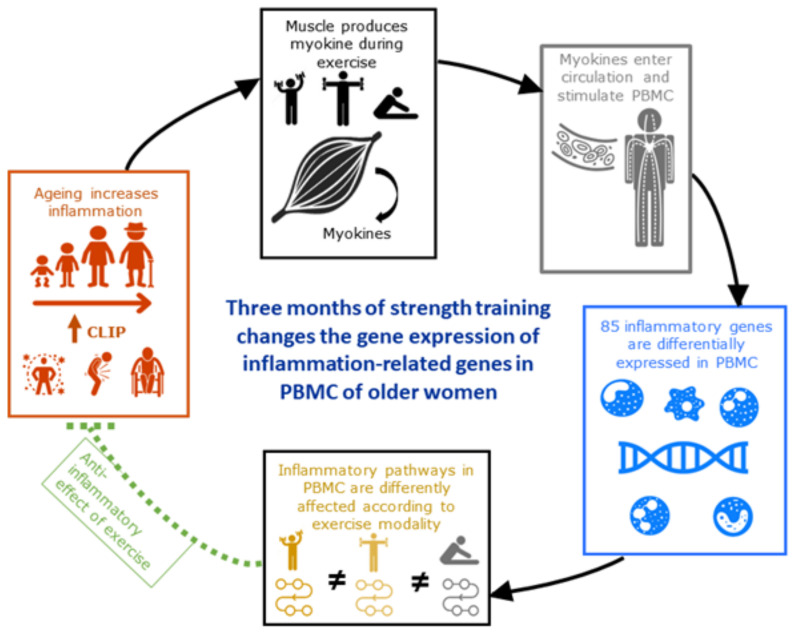
Impact of strength training on inflammatory gene expression in PBMC.

**Table 1 cells-11-00531-t001:** Baseline characteristics from the included women in the RNAseq experiment.

	IST *n* = 4	SET *n* = 5	CON *n* = 5	*p*-Value
**Weight (kg)**	62.62 ± 5.41	69.43 ± 12.52	64.17 ± 4.60	0.47
**Height (m)**	1.63 ± 0.05	1.59 ± 0.06	1.54 ± 0.06	0.11
**Body Mass Index (kg/m²)**	23.34 ± 1.86	28.50 ± 3.72	27.44 ±3.93	0.11
**Age (years)**	70.09 ± 4.64	72.13 ± 5.39	69.45 ± 2.62	0.61
**MMSE (score: 0–30)**	30.00 ± 0.00	27.2 ± 2.95	29.20 ± 0.84	0.10
**Exercise compliance (%)**	69.96 ± 19.47	93.41 ± 18.86	80.98 ± 7.31	0.13
**MSI (kg)**	24.34 ± 6.10	33.70 ± 7.71	37.61 ± 14.51	0.20
**Health categories (*n*)**				
**A**	0	1	0	0.62
**B**	4	3	5	
**C**	0	1	0	
**Charlson Index**	0.50 ± 1.00	0.00 ± 0.00	0.40 ± 0.89	0.58
**Medication use (*n*)**	4.25 ± 4.35	1.20 ± 1.9	4.40 ± 2.41	0.20
**Hypertension (*n*)**	0	1	2	0.35
**Diabetes (*n*)**	0	0	0	1.00
**YPAS-ADS (score 0–177)**	44.75 ± 9.18	32.00 ± 15.26	38.00 ± 27.29	0.63
**White blood cells (10^−3^/μL)**	5.90 (5.29–8.72)	6.51 (4.55–6.82)	6.28 (5.29–6.62)	0.57
**Neutrophils (%)**	50.75 (48.70–69.35)	57.42 (55.65–67.47)	43.44 (37.60–64.57)	0.94
**Monocytes (%)**	8.38 (7.91–10.08) †	5.45 (4.70–6.88)	7.81 (5.58–8.00)	0.02
**Eosinophils (%)**	1.43 (0.60–1.98)	2.77 (0.79–4.01)	3.99 (1.81–5.38)	0.19
**Basophils (%)**	0.37 (0.27–0.50)	0.58 (0.39–0.80)	0.79 (0.56–1.41)	0.05
**Lymphocytes (%)**	26.05 (21.65–38.02)	29.40 (26.90–33.20)	28.7 (24.20–43.62)	0.73

Data represent mean ± SD or median (min–max), * One-way ANOVA, Fischer’s exact test or Kruskall-Wallis test. † Significantly different from SET (Mann-Whitney U test *p* < 0.05). MMSE: mini mental state examination. MSI: muscle strength index. YPAS-ADS = Activity Dimensions Summary score of the Yale Physical Activity Survey.

## Data Availability

The datasets used and/or analysed during the current study are available from the corresponding author on reasonable request.

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
