# Peer review of "Three Months of Strength Training Changes the Gene Expression of Inflammation-Related Genes in PBMC of Older Women: A Randomized Controlled Trial"

_cells, 2022, doi:10.3390/cells11030531_

Round 1
Reviewer 1 Report
This is a very interesting study about the anti-inflammatory effects of strength training in older woman. The study was done through molecular expression of PBMC during the training period. The hypothesis concerning the link myokine and PBMC gene expression is here well introduced
I have two major comments
- The first one concerns the very poor number of subjects per group to have a strong conclusion in term of statistics.
- The second concerns the manuscript which is very rich... but to rich for an easy reading. I have the same comment for the abstract in which some abbreviations have not been defined: LXR/RRX … TREM1... So the abstract and the results sections have to be simplified. The discussion section should be shortened.
I have few minor comments
- Figure 1. Is it possible to add the BMI, the Charlson index and the number of daily drugs per group
- In the result section, the signaling presentations (TREM1, LXR/RXR for example) are more corresponding to a “discussion” approach than a “result” one. Same comment for the pathway "cytotoxic T..."
- Beside the different signaling TREM1, LXR/RXR, is there any modification of gene expression related to SASP?
Author Response
This is a very interesting study about the anti-inflammatory effects of strength training in older woman. The study was done through molecular expression of PBMC during the training period. The hypothesis concerning the link myokine and PBMC gene expression is here well introduced
I have two major comments
1. The first one concerns the very poor number of subjects per group to have a strong conclusion in term of statistics.
RESPONSE TO THE REVIEWER:
As mentioned in the statistical analysis section (see lines 272-274) we performed an a priori power calculation using Gpower®, showing that a sample size of 4 participants per group allowed to detect an exercise-induced gene expression fold change of at least 1.5 with a power of 0.82 and alpha=0.017. We added a comment to the strengths/limitations section on line 672-674.
2. The second concerns the manuscript which is very rich... but to rich for an easy reading. I have the same comment for the abstract in which some abbreviations have not been defined: LXR/RRX … TREM1... So the abstract and the results sections have to be simplified. The discussion section should be shortened.
RESPONSE TO THE REVIEWER:
We simplified the abstract and we added an explanation for the abbreviations of the pathways in the legend to figure 5. The abbreviations of the individual genes are explained in figures 2-4. We hope that this improved the readability of the results section.
I have few minor comments
1. Figure 1. Is it possible to add the BMI, the Charlson index and the number of daily drugs per group
RESPONSE TO THE REVIEWER:
We added the requested information to table 1.
2. In the result section, the signaling presentations (TREM1, LXR/RXR for example) are more corresponding to a “discussion” approach than a “result” one. Same comment for the pathway "cytotoxic T..."
RESPONSE TO THE REVIEWER:
We understand this comment. However, we didn’t mix results and discussion, but we added the relevant information embedded in supplementary figure 1 to the results section. This to provide a better understanding of that results section. In order to clarify this to the readers we added a comment explaining why the results regarding the pathways are reported in line with the affected corresponding genes. We also referred to supplementary figure 1 for more detailed information. See line 403-406.
3. Beside the different signaling TREM1, LXR/RXR, is there any modification of gene expression related to SASP?
RESPONSE TO THE REVIEWER:
Thank you for this comment. Indeed, many of the affected pathways are related to the Senescence Associated Secretory Profile (SASP) , such as TREM1, LXR/RXR, Toll-Like Receptor Signaling, T-cell Ex-haustion Signaling, Telomerase Signaling and Th2 pathways. This supports the hypothesis that strength training can affect and/or attenuate SASP, resulting in lower CLIP. We added a comment to the discussion section (see line 538-542).
Reviewer 2 Report
Manuscript ID: cells-1431508
Type of manuscript: Article
Title: Three months of strength training changes the gene expression of
inflammation-related genes in PBMC of older women: a randomized controlled
trial
This is a good manuscript that authors concluded three months IST and SET can both induce changes in CLIP-related gene 26 expression in PBMC, but by affecting different genes and related pathways.
Method is good. Results is Ok.
Some minor revision is needed.
- Use new Reference in introduction.
- Added ethical code in the M&M
Author Response
This is a good manuscript that authors concluded three months IST and SET can both induce changes in CLIP-related gene 26 expression in PBMC, but by affecting different genes and related pathways.
Method is good. Results is Ok.
Some minor revision is needed.
1. Use new Reference in introduction.
RESPONSE TO THE REVIEWER:
We added the following recent references to the introduction:
- Line 38 & line 41: “Rogeri, P.S., et al., Crosstalk Between Skeletal Muscle and Immune System: Which Roles Do IL-6 and Glutamine Play? Front Physiol, 2020. 11: p. 582258.”
- Line 88: “Andonian, B.J., et al., Altered skeletal muscle metabolic pathways, age, systemic inflammation, and low cardiorespiratory fitness associate with improvements in disease activity following high-intensity interval training in persons with rheumatoid arthritis. Arthritis Res Ther, 2021. 23(1): p. 187.”
- Line 93: “Hayes, L.D., et al., High Intensity Interval Training (HIIT) as a Potential Countermeasure for Phenotypic Characteristics of Sarcopenia: A Scoping Review. Front Physiol, 2021. 12: p. 715044.”
2. Added ethical code in the M&M
RESPONSE TO THE REVIEWER:
We added the RIB approval code on line 146.
Reviewer 3 Report
This manuscript aimed to investigate changes of inflammation-related gene-expression in peripheral mononuclear blood cells (PBMC) by strength training. The study is well designed and written, and presents novelty and interesting data, but some points needs to be clarify.
Introduction:
- Please explain better the difference in IL-6 release and action as pro or anti-inflammatory role in the first paragraph. The acute anti-inflammatory and chronic pro-inflammatory roles need to be better explained in the first paragraph.
- Please point out that this IL-6 responses are similar in all exercises ou only in resistance exercises.
- In the lines 79 to 86, the authors pointes out about the the dose-response relationship between different intensities of strength training, but to me is not clear what it is important to study this and why this differences happens. I suggest to improve this information to lead the readers better to the aim of the study.
Methods
- Please provide more information about the women that participated in the study. How many had hypertension, diabetes, or other diseases? What kind of medicine they use? What was their physical activity level and experience in strength training?
- How many times each participant performed in the end of the exercise training? Since they were asked to exercise 2-3 times weekly. Was it different in each group? If so, this could be listed as a limitation of the study.
- Did they performed a familiarization in the 1RM test? The literature shows that this is very important in this age population.
- Lines 164 and 165: "participants were asked to train 3x/week for 3 months resulting in a target number of 36 sessions". How many completed this entire training sessions?
Results/Discussion
- Please provide the initial number os volunteers and the dropout values in each group. How many were excluded? How many were screened ?
- I suggest to add the training results as main table/figure in the manuscript.
- The lack of difference in between IST and SET groups in strength and gene expression should be better explained in the discussion. What are the main difference between training intensity responses and why both strength and gene expression were similar? There are many studies in the literature showing that training intensities may have different physiological responses.
- Was the sample size in each group a significant limitation of the study?
- The study weakness and strength should be better explained. This paragraph is confuse.
Author Response
This manuscript aimed to investigate changes of inflammation-related gene-expression in peripheral mononuclear blood cells (PBMC) by strength training. The study is well designed and written, and presents novelty and interesting data, but some points needs to be clarify.
Introduction:
- Please explain better the difference in IL-6 release and action as pro or anti-inflammatory role in the first paragraph. The acute anti-inflammatory and chronic pro-inflammatory roles need to be better explained in the first paragraph.
RESPONSE TO THE REVIEWER:
We added an explanation regarding the difference between the exercise-induced myokine response and the cytokine release during inflammation (see line 49-57)
- Please point out that this IL-6 responses are similar in all exercises ou only in resistance exercises.
RESPONSE TO THE REVIEWER:
We have added information on the IL-6 responses following different types of exercise in the introduction section (see line 41-49).
- In the lines 79 to 86, the authors pointes out about the the dose-response relationship between different intensities of strength training, but to me is not clear what it is important to study this and why this differences happens. I suggest to improve this information to lead the readers better to the aim of the study.
RESPONSE TO THE REVIEWER:
We added additional information regarding the possible underlying mechanisms of the dose-response relationship and its importance for exercise prescription to counter CLIP in older persons (see line 105-109).
Methods
- Please provide more information about the women that participated in the study. How many had hypertension, diabetes, or other diseases? What kind of medicine they use? What was their physical activity level and experience in strength training?
RESPONSE TO THE REVIEWER:
We added the requested Information to table 1. Subjects performing currently or within the past 6months, on a regular basis, physical exercise at higher intensities than habitual daily activity (e.g. fitness classes, strengthening exercises, cycling club) were excluded. We added this information as well as a description how physical activity level was measured to the methods section (see line 126-128 and line 132-136).
- How many times each participant performed in the end of the exercise training? Since they were asked to exercise 2-3 times weekly. Was it different in each group? If so, this could be listed as a limitation of the study.
RESPONSE TO THE REVIEWER:
Exercise compliance was calculated as follows: participants were asked to train 3x/week for 3 months resulting in a target number of 36 sessions. The effectively performed number of training sessions was recorded and was expressed as a percentage of the target number (see line 192-195). We found no significant difference in compliance between the three groups (see table 1). To improve the readability we changed the variable name in table 1 to “Exercise compliance” and added also a few words in the results section (see line 306-307).
- Did they performed a familiarization in the 1RM test? The literature shows that this is very important in this age population.
RESPONSE TO THE REVIEWER:
Yes, the accommodation period and 1RM procedure are described in detail on line 174-184.
- Lines 164 and 165: "participants were asked to train 3x/week for 3 months resulting in a target number of 36 sessions". How many completed this entire training sessions?
RESPONSE TO THE REVIEWER:
All participants ended the 3-month training period. Exercise compliance was calculated as the effectively performed number of training sessions expressed as a percentage of the target number (i.e. 36 sessions, see 192-195).
Results/Discussion
- Please provide the initial number os volunteers and the dropout values in each group. How many were excluded? How many were screened ?
RESPONSE TO THE REVIEWER:
As mentioned on line 115-118 and line 137-143, this is a sub-study of a larger, ongoing, RCT. For this sub-study, similarly as in our previously published paper (Cao Dinh, H., et al., J Gerontol A Biol Sci Med Sci, 2019. 74(12): p. 1870-1878.), we included only women as the target sample size for the female participants has been reached. Women who completed 3 months of intervention, and did not become ill during the training period, and did not change their medication intake for at least 6 weeks prior blood sampling, and were sufficiently compliant to the intervention, and of which the RNA samples had an RNA Quality Number (RQN) higher than 7 and a yield higher than 8 ng/µl were eligible for further analysis.
- I suggest to add the training results as main table/figure in the manuscript.
RESPONSE TO THE REVIEWER:
We added a new figure 1 showing the changes in muscle strength after 3 months intervention. The other figures were renumbered accordingly.
- The lack of difference in between IST and SET groups in strength and gene expression should be better explained in the discussion. What are the main difference between training intensity responses and why both strength and gene expression were similar? There are many studies in the literature showing that training intensities may have different physiological responses.
RESPONSE TO THE REVIEWER:
As mentioned on line 530-532, IST and SET altered only 14 genes in similar direction whereas 19 genes were altered in opposite direction. We added a comment on line 532-533 to stress the differences in gene expression between IST and SET, despite the comparable strength gains.
- Was the sample size in each group a significant limitation of the study?
RESPONSE TO THE REVIEWER:
As mentioned in the statistical analysis section (see lines 272-274) we performed an a priori power calculation using Gpower®, showing that a sample size of 4 participants per group allowed to detect an exercise-induced gene expression fold change of at least 1.5 with a power of 0.82 and alpha=0.017. We added a comment to the strengths/limitations section on line 672-674.
- The study weakness and strength should be better explained. This paragraph is confuse.
RESPONSE TO THE REVIEWER:
We restructured the strengths/limitation part of the discussion section (see line 636-679) in order to improve the readability.
Round 2
Reviewer 3 Report
All changes improved the manuscript, so now it is appropriated to be published.